# A Case of Bullous Pemphigoid with Significant Infiltration of CD4-Positive T Cells during Treatment with Pembrolizumab, Accompanied by Pembrolizumab-Induced Multi-Organ Dysfunction

**DOI:** 10.3390/diagnostics14171958

**Published:** 2024-09-04

**Authors:** Yoshihito Mima, Tsutomu Ohtsuka, Ippei Ebato, Yoshimasa Nakazato, Yuta Norimatsu

**Affiliations:** 1Department of Dermatology, Tokyo Metropolitan Police Hospital, Tokyo 164-8541, Japan; 2Department of Dermatology, International University of Health and Welfare Hospital, Tochigi 324-8501, Japan; 3Department of Diagnostic Pathology, International University of Health and Welfare Hospital, Tochigi 324-8501, Japan; 4Department of Dermatology, International University of Health and Welfare Narita Hospital, Chiba 286-0124, Japan; norimanorima@gmail.com

**Keywords:** PD-1 checkpoint inhibitors, bullous pemphigoid, diarrhea, hypothyroidism, multi-organ immune-related adverse events, CD4-positive T cells

## Abstract

Immune checkpoint inhibitors (ICIs) activate T cells, causing immune-related adverse events (irAEs). Skin manifestations are common among irAEs, but ICI-associated bullous pemphigoid (BP) is rare. Inhibiting programmed death (PD)-1 signaling, in addition to causing epitope spreading, may disrupt B and T cell balance, causing excessive autoantibody production against the skin’s basement membrane, leading to BP. A 70-year-old woman developed late-onset multi-organ irAEs, including diarrhea, thyroid dysfunction, and BP, while receiving pembrolizumab, a PD-1 inhibitor. This highlights the long-term risk of irAEs, which can occur 2–3 years after starting ICIs. In cases of multi-organ irAE, C-reactive protein levels and neutrophil/lymphocyte ratio are often low. These characteristics were observed in our case. Few papers address multiple organ involvement, highlighting the need to consider irAEs in a multi-organ context. While it is known that drug-induced skin reactions worsen as blood eosinophil counts increase, in our case, the eosinophil count remained normal, suggesting that ICI-associated BP might have been controlled without discontinuing the ICI and through tapering of low-dose oral prednisone treatment. Additionally, in this case, significant CD4-positive T cell infiltration was observed in the immunostaining examination of the blisters, indicating that severe CD4-positive T cell infiltration induced by the ICI might have led to multi-organ involvement, including severe diarrhea. Few reports focus on blood eosinophil counts in BP cases or discuss CD4 and CD8 immunostaining in BP cases. Therefore, future research should explore the relationship between blood eosinophil counts, immunostaining results, and the prognosis of irAEs, including BP, in treatment courses.

## 1. Introduction

Immune checkpoints are mechanisms through which cancer cells disguise themselves to evade immune surveillance and avoid attacks within the body [1]. They function as negative regulators of the immune system, mediate self-tolerance, prevent autoimmunity, and protect tissues from immune attacks [1]. Tumor cells enhance immune resistance by activating specific immune checkpoint proteins, thereby inhibiting T cell activation [2].

The primary immune checkpoints include Programmed Death-1 (PD-1) and its ligand PD-L1 (Programmed Death Ligand 1), as well as Cytotoxic T-Lymphocyte Antigen 4 (CTLA-4) [3]. PD-1 (CD279), a member of the CD28 family, is a co-inhibitory transmembrane protein expressed on antigen-stimulated T cells, B cells, natural killer cells (NK), and myeloid-derived suppressor cells [4]. Upon binding to its ligand, PD-1 reduces T cell responses to T cell receptor stimulation, thereby modulating the intensity of the immune response [4]. PD-L1 is expressed by tumor cells, epithelial cells, dendritic cells, macrophages, fibroblasts, and exhausted T cells [5]. Cytokines such as interferon-gamma and carcinogenic factors influence the intensity of its expression [5]. When PD-L1 binds to PD-1, it suppresses the PI3K-AKT and Ras-Raf-MEK-ERK signaling pathways, thereby inhibiting the proliferation and differentiation of effector T cells [5]. CTLA-4 is a type I transmembrane glycoprotein of the immunoglobulin superfamily that is highly expressed in tumor tissues and is generally present in the cytoplasm of CD4 and CD8-positive T cells [6]. It is considered a negative regulator of antitumor immunity, as it binds with high affinity to CD80 and CD86 on the surface of antigen-presenting cells. This interaction inhibits cytotoxic T-cell activity and enhances regulatory T cell (Treg) immunosuppressive activity, facilitating tumor immune evasion [6]. Thus, PD-1, PD-L1, and CTLA-4 act to diminish T cell responses and inhibit T cell functions [3,4,5,6].

Immune checkpoint blockade disrupts the inhibitory pathways that naturally constrain T cell reactivity, thereby limiting the activation and maintenance of T cell effector functions [7]. Immune checkpoint inhibitors (ICIs), such as anti-CTLA-4 and anti-PD-1/PD-L1 monoclonal antibodies, have demonstrated robust antitumor activity in malignancies such as urothelial carcinoma, renal cell carcinoma, melanoma, non-small cell lung cancer, colorectal cancer, and Hodgkin lymphoma and are widely used in clinical practice [8]. Commonly used ICIs include anti-PD-1/PD-L1 monoclonal antibodies, such as nivolumab, pembrolizumab, atezolizumab, durvalumab, avelumab, and cemiplimab, and anti-CTLA-4 monoclonal antibodies, such as ipilimumab and tremelimumab [8].

ICIs disrupt the body’s immune balance, leading to the development of a series of immune-related adverse events (irAEs) [9]. Although the precise pathophysiological mechanisms underlying irAEs remain unclear, they share several similarities with other autoimmune diseases. IrAEs are believed to be associated with changes in the functions of the body’s autoimmune system, such as the breakdown of self-tolerance and increased sensitivity to antigen recognition, resulting in attacks on the body’s tissues [10].

The normal function of immune regulation in the body is to balance immune activation and tolerance through the co-stimulatory pathways of reactive T cells [11]. Immune tolerance suppresses the activation of autoreactive T cells and regulates the strength of the immune system [11]. Inhibitory costimulatory molecules on naïve T cells bind to T cell ligands, modulating the balance between T cell activation, tolerance, and immune-mediated tissue damage [11].

ICIs activate not only CD4-positive T cells but also CD8-positive T cells. They inhibit immune checkpoint molecules to prevent tumor cell immune escape and disrupt peripheral T cell tolerance. This leads to the rapid diversification and clonal expansion of toxic cells, resulting in heightened inflammation and autoimmunity [12,13,14]. Thus, ICIs can potentially cause irAEs through their effects on CD4+ T cells, CD8+ T cells, and Tregs. 

Increased activation of auto-reactive B cells and the production of autoantibodies may cause damage by binding to target antigens, triggering classical complement cascade reactions [15]. 

Significant changes in cytokines, such as interleukins (ILs), tumor necrosis factors, and interferons, before and after treatment may play a role in patients with irAEs. The release of inflammatory mediators by immune cells can cause immune-mediated damage to tissues with anatomical susceptibility, suggesting that tissue-specific or general cytokine levels are involved in irAE pathogenesis [16]. These cytokines bind to immune cells and activate intracellular signaling pathways, such as the Janus kinase-signal transducers and activators of transcription pathways, causing the dysregulation of inflammatory responses [16].

Thus, many factors, including T cell infiltration, autoantibody production, and the mediation of inflammatory cytokines such as ILs, are believed to be deeply intertwined, leading to the development of organ-specific irAEs [11,12,13,14,15,16].

ICI-related irAEs are organ-specific, with skin-related irAEs being the most common (especially mild pruritus and rash), followed by gastrointestinal toxicities, often presenting as diarrhea or colitis [17]. Endocrine irAEs are the third most common, including thyroid dysfunction, hypothyroidism, hyperthyroidism, pituitary inflammation, and adrenal insufficiency [17]. Musculoskeletal toxicities such as mild arthralgia and myalgia and ocular toxicities such as mild dry eye syndrome and uveitis have also been frequently reported [17]. Although less common, pneumonitis, myocarditis, neurotoxicity, myositis, nephritis, and hematological toxicities may be severe [17].

Despite the high frequency of skin irAEs, two-thirds of patients with skin irAEs require systemic corticosteroids for rash treatment, and 19% of patients discontinue cancer immunotherapy owing to irAEs [18]. Common mild-to-moderate skin irAEs include pruritus, non-specific maculopapular rash, lichen planus-like eruptions, psoriasis-like dermatitis, and eczematous eruptions with vitiligo and alopecia. Severe skin irAEs, such as bullous pemphigoid (BP), Stevens–Johnson syndrome, toxic epidermal necrolysis, and drug-induced hypersensitivity syndrome/drug reaction with eosinophilia and systemic symptoms, have also been reported [18].

BP, which has garnered attention as a skin manifestation of irAEs, is a common autoimmune blistering skin disease characterized by itching, urticaria, or eczematous eruptions, followed in most cases by blister formation [19]. However, some patients present with morphologically atypical variations in BP, making the diagnosis challenging [19]. Hemidesmosomal proteins BP180 (BP antigen 2 and collagen XVII) and BP230 (BP antigen 1) have been identified as the main autoantigens involved in the pathogenesis of BP [20]. Autoantibodies against these proteins react, leading to neutrophil and complement pathway activation, which damages the basement membrane, resulting in BP [21]. Although adverse skin reactions are frequently observed after ICI administration, the occurrence of BP due to ICI treatment is rare [19].

Herein, we report a rare case of BP that developed during ICI therapy and was accompanied by multi-organ irAEs, including thyroid dysfunction and diarrhea. 

## 2. Case Presentation

A 70-year-old Japanese woman was found to have a black lesion in her stomach during routine upper gastrointestinal endoscopy 3 years prior to her first dermatological consultation (Figure 1). 

A biopsy of the black lesion detected during endoscopy revealed the proliferation of atypical cells containing melanin upon hematoxylin and eosin (HE) staining (Figure 2a). The atypical cells tested positive for melan-A staining (Figure 2b). These findings led to the diagnosis of malignant melanoma of the gastric mucosa. 

A computed tomography (CT) scan performed from the head to the pelvis to search for metastases revealed a right lung nodule, leading to a diagnosis of lung metastasis (Figure 3a). After endoscopic surgery for gastric mucosal melanoma, pembrolizumab, an ICI, was initiated at 200 mg every 3 weeks. Four months after initiating ICI therapy, the patient developed hypothyroidism, which was considered an irAE, and thyroid hormone medication was administered. Following treatment, her thyroid function normalized, allowing continuation of ICI therapy. For the next 2 years, no new irAEs appeared, and the lung metastatic nodule showed a significant reduction (Figure 3b). 

Three years after starting ICI therapy, the patient developed widespread rashes that did not improve after 3 weeks of a strong-class topical steroid and was referred to our dermatology department. Physical examination revealed multiple edematous erythematous lesions on the trunk and extremities, ranging in size from the thenar eminence to a hen’s egg (Figure 4a). Despite upgrading the steroid to the strongest class and treating for 2 weeks, the erythema continued to expand and new small blisters appeared (Figure 4b). 

Blood tests showed normal eosinophil levels (417/μL) but elevated immunoglobulin (Ig) E levels (516 IU/mL). Antibody tests for bullous diseases were negative for anti-desmoglein 1 and 3 antibodies but positive for anti-BP180 antibodies (216). Histopathological examination of a biopsy specimen from the blistered area revealed subepidermal blisters with eosinophils and lymphocytes infiltrating the small blisters along with marked infiltration of inflammatory cells into the upper dermis (Figure 5a,b). 

Immunofluorescence staining of a biopsy specimen from the erythematous area showed no IgA, IgM, or C4 deposition but showed linear deposition of IgG and C3 along the basement membrane (Figure 6a,b).

Immunostaining of the blistered area revealed that the infiltrating lymphocytes in the dermis were mainly CD4-positive T cells (Figure 7a). Few CD8-positive T cells were observed in the upper dermis, and granzyme B staining was positive (Figure 7b). No CD20-positive B-cells were observed in the dermis (Figure 7c). These findings led to the diagnosis of BP associated with ICI therapy.

Treatment was initiated with oral prednisolone (20 mg), the strongest topical steroid. After 1 month, no new blisters had formed, and the edematous erythema began to subside, allowing for gradual tapering of oral corticosteroids. Four months after starting steroids, diarrheal symptoms appeared when the prednisolone dose was reduced to 10 mg and worsened significantly, leading to a temporary increase in the prednisolone dose to 12 mg and discontinuation of pembrolizumab. After the diarrheal symptoms resolved, prednisolone tapering was resumed, and the dose was reduced to 7 mg without any recurrence of diarrhea symptoms. Regarding BP, no new blisters appeared, although pale erythema persisted. Six months after discontinuing pembrolizumab, the lung metastasis has not exacerbated.

## 3. Discussion

BP is a rare skin-related irAE that is associated with ICIs [19]. Carlos G et al. reported the first case of bullous pemphigoid in a patient with metastatic melanoma treated with pembrolizumab in 2015 [22]. Since then, the number of case reports discussing ICIs-induced BP has grown, leading to a substantial body of case series and review articles on the subject [23,24]. Many patients develop nonspecific skin rashes with pruritic eczematous dermatitis, including papules or plaques, before the appearance of blisters [24]. In our case, erythema resembling lichen planus appeared extensively before the BP blisters developed. Approximately 0.3% of patients develop BP during ICI therapy [24]. A report summarizing 58 patients who developed BP while on ICI therapy indicated that the median time from the start of treatment to the first skin lesions was 21 weeks (range: 1–88 weeks), with 51 patients (87.9%) developing blisters [25]. The median time to blister formation was 27.5 weeks (range: 3–104 weeks) after the initiation of anti-PD-1/PD-L1 therapy [25]. In contrast, Kawsar et al.’s study of 16 cases reported a median onset of BP at approximately 50 weeks from the start of ICI therapy, suggesting significant variability in BP onset among different case series [24]. In our case, BP developed three years after starting ICI therapy, which is later than the average onset of 7–12 months. However, cases have been reported wherein BP appeared 2–3 years after ICI therapy, indicating onset variability between individual cases [24,25]. Additionally, some patients developed blisters after discontinuing PD-1/PD-L1 inhibitors [24].

In our case, typical findings of BP were observed, including subepidermal blisters, eosinophilic infiltration, and linear deposition of IgG and C3 along the dermoepidermal junction (DEJ). The case series observed subepidermal blisters and eosinophils in 70.7% (41/58) and 81% (47/58) of patients, respectively, and linear deposition of IgG and/or C3 along the DEJ was observed in 86.8% (46/53) of cases [24]. However, cases of BP have been reported without typical pathological features [24]. 

BP is an autoimmune skin disease wherein the immune system mistakenly attacks proteins in the skin, causing blisters and skin detachment [21]. BP230, a hemidesmosomal protein, anchors skin cells to the basement membrane and stabilizes keratin filaments. BP230 becomes a target of autoimmune responses, recognized by autoantibodies present in the blood of patients with BP [26,27]. BP180, a transmembrane protein, extends partially outside the cell, making it susceptible to autoantibody binding. BP180 also plays a role in anchoring cells to the basement membrane, with the NC16A domain being the major epitope [28]. In BP, IgG antibodies bind to the basement membrane and activate the complement system, which, in turn, activates mast cells and attracts neutrophils [29]. Neutrophils release elastase, which disrupts the basement membrane and results in blister formation [30,31]. Although complement activation is important, it is not always necessary for disease progression. Many BP patients also have IgE antibodies, which contribute to immune responses by binding to mast cells and basophils, further activating eosinophils [32,33].

Regarding BP associated with anti-PD-1/PD-L1 therapy, the spread of epitopes is observed, where the immune response extends to additional targets [34]. This phenomenon, which is also observed with dipeptidyl peptidase-4 inhibitors, results in a broader immune response [35]. Anti-PD-1/PD-L1-induced autoimmune phenomena are thought to result from dysregulation of B and T cells [34]. PD-1 signaling in B cells inhibits excessive proliferation by preventing tumor antigens, such as BP180, from binding to B cell receptors (BCR) [36]. Blocking PD-1 enhances BCR responses, promoting B-cell proliferation and antibody production, leading to cross-reactive immune responses against the skin’s basement membrane [36].

PD-1 signaling also affects T cell-dependent humoral immunity, in which interactions between follicular helper T cells (Tfhs) and follicular regulatory T cells (Tfrs) regulate B cell function [37]. Tfhs aid in the selection and maintenance of B cells and promote their differentiation into antibody-producing plasma or memory B cells, whereas Tfrs suppress excessive immune responses [37]. Blocking PD-1 disrupts this balance, leading to increased production of low-affinity plasma cells and antibodies, potentially causing antibody-mediated autoimmune phenomena such as BP [34,37].

Th2-related cytokines, including IL-4, IL-5, IL-6, IL-10, and IL-13, are involved in BP pathogenesis [38]. Increased expression of Th2 cytokines has been observed in the serum, skin biopsies, and blister fluid, indicating their close association with BP development [38]. Dupilumab, which targets IL-4 and IL-13, has shown promising results in preliminary trials of BP, suggesting a strong link to Th2-type inflammation [39]. PD-L2 activation is related to Th2 inflammation, and PD-1 inhibitors may disrupt the PD-1/PD-L2 pathway, thereby promoting Th2-type inflammation and BP development [38,40]. CD4 + T cells, particularly Th2 cells, play a significant role in BP by interacting with B cells through CD40-CD40L interactions, leading to B cell differentiation and epitope spreading of IgG and IgE autoantibodies [41]. In our case, significant CD4-positive T cell infiltration suggested a contribution to the development of BP.

Kridin et al. conducted a large population-based longitudinal study on the association between melanoma and BP and reported a 50% increased risk of subsequent BP in patients with a history of melanoma [42]. BP180’s 60 kDa endodomain is expressed in malignant melanoma but not in benign melanocytic tumors, suggesting its role in malignant transformation [43]. Cases of BP developing during ICI therapy for melanoma are often classified as ICI-associated BP [24,25]. Distinguishing between melanoma- and ICI-induced BP can be challenging, with both potentially contributing to its development.

Marques-Piubelli ML et al. reported significantly lower CD20-positive B cell densities in irAE-BP samples compared to de novo BP samples [44]. In our case, no B cell infiltration into the dermis or the presence of cytotoxic CD8-positive cells supported the possibility of ICI-induced BP. A case series of 58 patients with BP after ICI therapy reported other organ irAEs, including joint pain (2 cases) and hypothyroidism (1 case), indicating that multi-organ irAEs are rare [45]. Most reports mention multi-organ involvement as supplementary information, with only one known report aggregating such cases [46].

Kichenadasse et al. investigated multi-organ irAEs in 1548 non-small cell lung cancer patients treated with the PD-L1 inhibitor atezolizumab [46]. They reported 730 irAEs in 424 patients (27%), with skin irAEs being the most common (42%), followed by laboratory abnormalities (27%), endocrine disorders (11.6%), neurological disorders (7.6%), and pulmonary (6.2%) irAEs. Multi-organ irAEs were observed in 84 patients (5.4%), with 70 having two-organ irAEs, 13 having three-organ irAEs, and 1 having four-organ irAEs [46]. In our case, multi-organ irAEs included BP, hypothyroidism, and diarrhea. Kichenadasse et al. reported 14 cases (0.9%) of irAEs involving three or more organs, indicating its rarity [46]. However, their study focused on anti-PD-L1 therapy, and different results may have been obtained with anti-PD-1 therapy [46]. Specific studies on multi-organ irAEs associated with anti-PD-1 therapy are lacking, warranting further research. Patients with multi-organ irAEs are more likely to be Caucasian and have a favorable performance status, low C-reactive protein levels, and a low baseline neutrophil-to-lymphocyte ratio. Multi-organ irAEs are associated with an improvement in overall survival, with a 50% survival period of over 2 years for patients with multi-organ irAEs, compared to 20 months for single-organ irAEs and 10 months for those without irAEs. In our case, the patient has been progression-free for 4 years since starting ICI therapy, suggesting a correlation between multi-organ irAEs and favorable prognosis.

Drago et al. reported that in drug-induced skin reactions, the eosinophil count typically increases as the skin symptoms become severe [47]. In drug eruptions, the higher the peripheral eosinophil count is, the more severe the clinical manifestations are, often requiring systemic treatment [47]. Moreover, these patients tend to have a longer recovery time compared to those without eosinophilia [48]. In practice, controlling ICI-associated BP is often challenging, typically necessitating the discontinuation of ICI therapy and the administration of high-dose prednisolone [23]. However, in the present case, the patient’s eosinophil count remained normal, which might have allowed for successful control of ICI-associated BP with the initiation of low-dose prednisone therapy without discontinuing ICI therapy. After the initiation of oral prednisolone treatment, eosinophil levels further decreased due to its effect, and subsequent measurements were not conducted. As a result, the eosinophil count during the acute exacerbation of irAE-related diarrhea could not be monitored in this case. Currently, there are no studies that have specifically examined the relationship between blood eosinophil levels and treatment outcomes in ICI-associated BP, indicating a need for further research in this area.

After developing hypothyroidism as an irAE, our patient developed BP. Additionally, during prednisolone tapering, new diarrheal symptoms emerged and gradually worsened, leading to ICI discontinuation. This case involved multi-organ irAEs with hypothyroidism, BP, and diarrhea. Earland et al. reported that an increased number of activated CD4-positive T cells is associated with severe irAEs, regardless of the organ affected [48]. Only one report on ICI-induced BP has discussed CD4 and CD8 immunostaining [49]. Our case showed a greater severity of CD4 infiltration than previously reported, suggesting that the severity of CD4 infiltration may be linked to multi-organ irAEs, including severe diarrhea. Although reports on CD4 and CD8 immunostaining in BP cases are limited, conducting such immunohistochemistry might help predict the severity of irAEs and the development of new multi-organ irAEs, such as diarrhea, during future treatment. 

Further accumulation of cases and research is needed to understand the mechanism linking the treatment outcomes of ICI-associated BP with peripheral eosinophil count and the degree of CD4 and CD8-positive T cell infiltration in the cutaneous tissue.

## Figures and Tables

**Figure 1 diagnostics-14-01958-f001:**
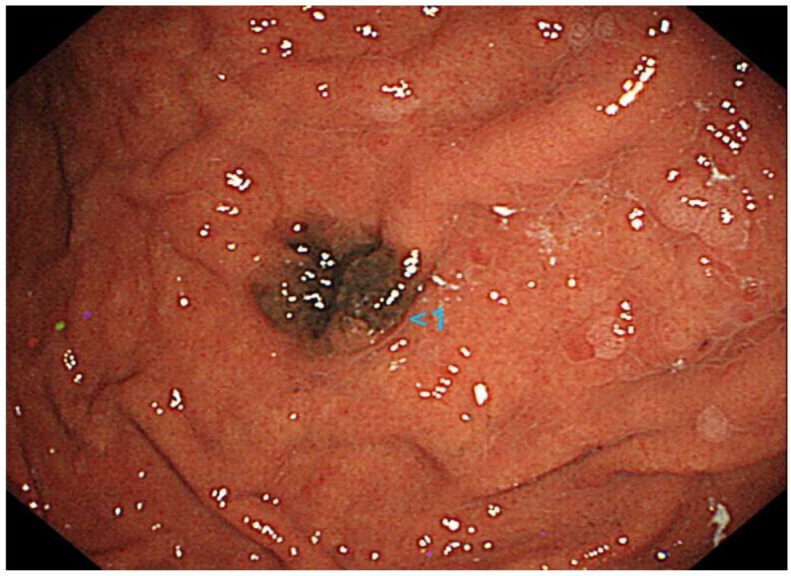
Upper gastrointestinal endoscopy reveals a black lesion in the stomach.

**Figure 2 diagnostics-14-01958-f002:**
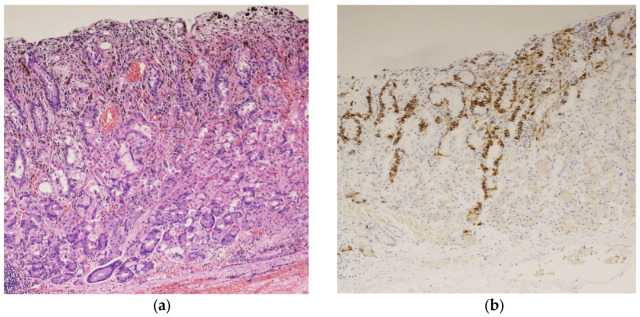
Histopathological examination of the black lesion reveals proliferation of atypical cells containing melanin pigment (hematoxylin and eosin staining; ×100) (**a**); the atypical cells tested positive for Melan-A staining (×100) (**b**).

**Figure 3 diagnostics-14-01958-f003:**
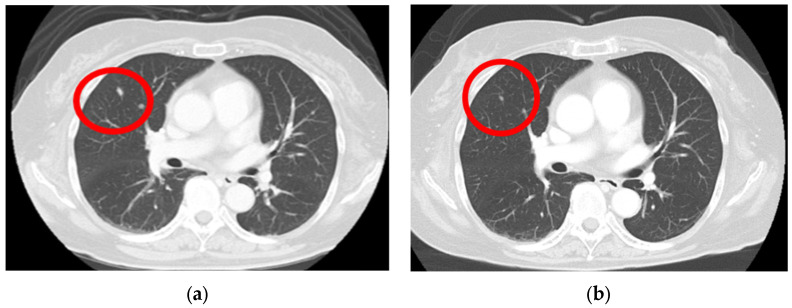
Lung metastasis is observed before the initiation of immune checkpoint inhibitors (**a**) (red circle); after the 3-year chemotherapy treatment, the lung metastasis nodule shows a significant regression (**b**) (red circle).

**Figure 4 diagnostics-14-01958-f004:**
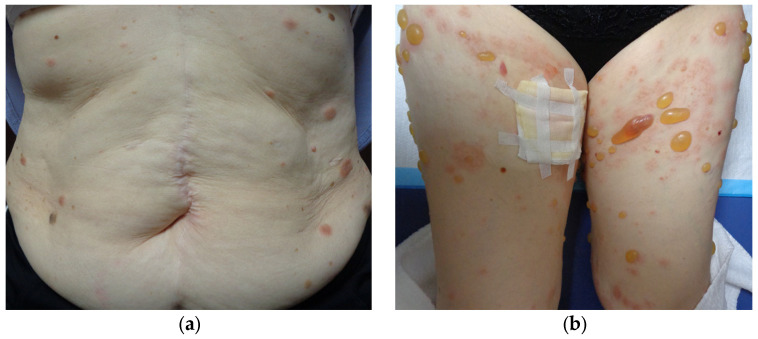
Multiple edematous erythematous lesions on the trunk and extremities, ranging in size from the thenar eminence to a hen’s egg, are observed (**a**); 2 weeks later, new small blisters are developed (**b**).

**Figure 5 diagnostics-14-01958-f005:**
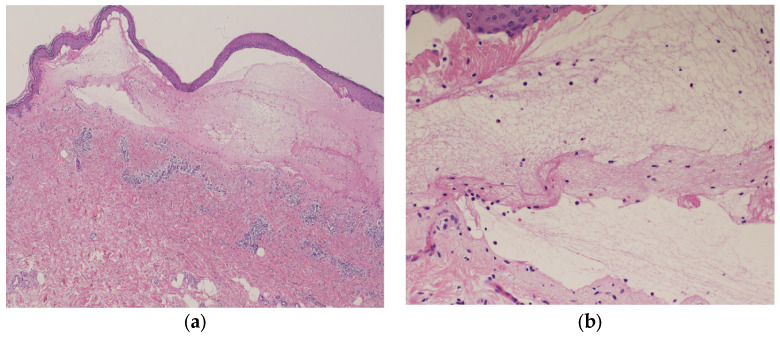
Histopathological examination of the blistered area reveals subepidermal blisters (hematoxylin and eosin [HE] staining; ×40) (**a**); eosinophils and lymphocytes were infiltrating around the small blisters, along with marked infiltration of inflammatory cells in the upper dermis (HE staining; ×200) (**b**).

**Figure 6 diagnostics-14-01958-f006:**
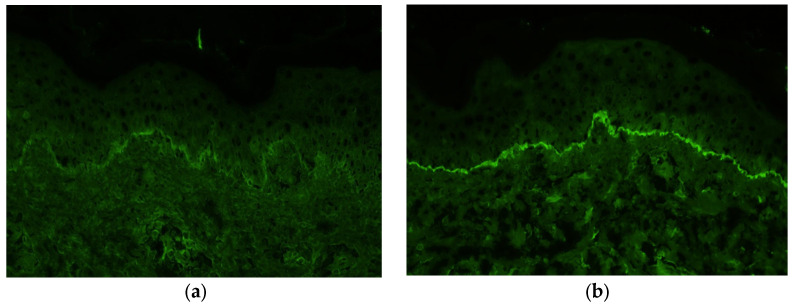
Immunofluorescence staining of a biopsy from the erythema shows deposition of immunoglobulin G (**a**) and C3 (**b**) along the basement membrane.

**Figure 7 diagnostics-14-01958-f007:**
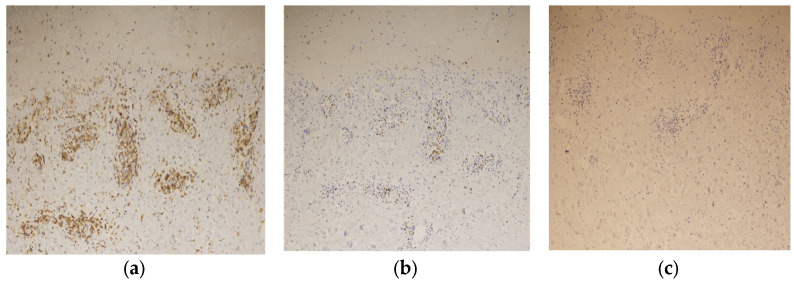
Immunostaining of the blister shows that the infiltrating lymphocytes are mainly positive for CD4 staining (×200) (**a**), slightly positive for CD8 staining (×200) (**b**), and negative for CD20 staining (×200) (**c**).

## Data Availability

Data concerning this article may be requested from the corresponding author for reasonable reasons.

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
