# Peer review of "A Case of Bullous Pemphigoid with Significant Infiltration of CD4-Positive T Cells during Treatment with Pembrolizumab, Accompanied by Pembrolizumab-Induced Multi-Organ Dysfunction"

_diagnostics, 2024, doi:10.3390/diagnostics14171958_

Round 1
Reviewer 1 Report
Comments and Suggestions for Authors
The manuscript clearly and convincingly presents a clinical case report describes unusual adverse reaction during pembrolizumab anti-cancer treatment of lung metastasis come from melanoma formerly found in stomach. This adverse effects appears as skin manifestation of bullous pemphigoid. The authors are thoroughly studied this adverse reaction, and evidently proved the diagnosis of bullous pemphigoid by histopathological examination and immunofluorescence staining of skin biopsy sample. The authors offer to a reader comprehensive introduction explaining possible immune system dysregulation leading to such autoimmune disease upon long term treatment of immune checkpoint inhibitors (ICI). At last, but not the least, is the correct patient's care with prednisolon to avoid adverse skin manifestations of ICI simultaneousely reducing the metastasis of the melanoma. I have no suggestions for the authors for the improving of this case report.
Author Response
Comments: The manuscript clearly and convincingly presents a clinical case report describes unusual adverse reaction during pembrolizumab anti-cancer treatment of lung metastasis come from melanoma formerly found in stomach. This adverse effects appears as skin manifestation of bullous pemphigoid. The authors are thoroughly studied this adverse reaction, and evidently proved the diagnosis of bullous pemphigoid by histopathological examination and immunofluorescence staining of skin biopsy sample. The authors offer to a reader comprehensive introduction explaining possible immune system dysregulation leading to such autoimmune disease upon long term treatment of immune checkpoint inhibitors (ICI). At last, but not the least, is the correct patient's care with prednisolon to avoid adverse skin manifestations of ICI simultaneousely reducing the metastasis of the melanoma. I have no suggestions for the authors for the improving of this case report.
Response:
Thank you very much for your thorough and detailed review.
Reviewer 2 Report
Comments and Suggestions for Authors
The authors prepared a well-written case report of BP developing during pembrolizumab treatment. The potential immunopathogenesis is thoroughly presented. The photographic documentation is of high quality.
In the Discussion section, I suggest to prepare a table summarizing published so far cases of BP during therapy with PD-1/PD-L1 inhibitors and its main features.
Author Response
Comments:
The authors prepared a well-written case report of BP developing during pembrolizumab treatment. The potential immunopathogenesis is thoroughly presented. The photographic documentation is of high quality.
In the Discussion section, I suggest to prepare a table summarizing published so far cases of BP during therapy with PD-1/PD-L1 inhibitors and its main features.
Response:
Thank you for your kind and thorough review. We appreciate your suggestions. We also considered summarizing all the reported cases; however, since the first case of ICI-BP was reported in 2015, more than several hundred cases have been documented. ICI-BP with multi-organ involvement, like in our case, is extremely rare. However, cases of ICI-BP without multi-organ involvement have become more common, and many of these may not be reported in the literature. Therefore, we thought that summarizing all cases would be impossible. We apologize for not being able to accommodate your suggestion. Instead, we added a section noting that since the first case was reported in 2015, numerous case series and reviews have been published, line 233-237. Thank you very much.